# Onset of Action of Selected Second-Generation Antipsychotics (Pines)–A Systematic Review and Meta-Analyses

**DOI:** 10.3390/biomedicines11010082

**Published:** 2022-12-29

**Authors:** Rikke Meyer, Kenneth Skov, Inderjeet Kaur Dhillon, Emilie Olsson, Niels Albert Graudal, Lone Baandrup, Gesche Jürgens

**Affiliations:** 1Clinical Pharmacology Unit, Zealand University Hospital, Roskilde, Sygehusvej 10, 4000 Roskilde, Denmark; 2Department of Clinical Medicine, University of Copenhagen, Blegdamsvej 3B, 2200 Copenhagen, Denmark; 3Mental Health Services West, Psychiatry Region Zealand, 4200 Slagelse, Denmark; 4Lupus and Vasculitis Clinic VRR4242, Copenhagen University Hospital, Blegdamsvej 9, 2100 Copenhagen, Denmark; 5Mental Health Center Copenhagen, Gentofte Hospitalsvej 15, Hellerup, 2900 Copenhagen, Denmark

**Keywords:** antipsychotic action, antipsychotic effect, antipsychotic agents, pines, psychopharmacology, schizophrenia, systematic review, meta-analysis

## Abstract

Recommendations for duration of treatment with antipsychotics before considering a switch vary from 2 to 8 weeks, although several studies suggest a rapid onset of action. The objective of this review was to estimate time to onset of action and time to maximum antipsychotic effect of asenapine, olanzapine, quetiapine, and zotepine (pines). We searched bibliographic databases for randomized, placebo-controlled trials in adults with schizophrenia estimating the antipsychotic effect of pines over time. Thirty-five studies including 6331 patients diagnosed with chronic schizophrenia were included. We estimated the standardized mean differences (SMD) of changes in symptom score from baseline to follow-up between intervention and placebo groups across studies using meta-analysis techniques. The summarized effect across all included pines administered as immediate-release formulations showed a statistically significant effect at week 1 (SMD, −0.20 [CI95% −0.28, −0.13]), which increased until week 3 (SMD, −0.42 [CI95% −0.50, −0.34]), after which the effect leveled off (week 6: SMD, −0.53 [CI95% −0.62, −0.44]). The sensitivity analyses of the individual pines confirm this finding, although data sparsity increases variability and limits conclusiveness of these analyses.

## 1. Introduction

Schizophrenia affects approximately 20 million people worldwide [1]. Untreated schizophrenia can be severely disabling and impair daily functioning.

Due to a more favorable side-effect profile, treatment guidelines for schizophrenia recommend second-generation antipsychotics (SGAs) as treatment of choice.

Pines are multi-acting receptor-targeted antipsychotics (MARTA) with an affinity for 5-TH2A-, D2-, cholinergic, and histaminergic receptors [2]. Small differences in receptor targeting and affinities between pines, however, lead to gradual differences in side effects. While all pines are sedatives due to anti-adrenergic and anti-histaminergic effects, high affinity for the H1-receptor makes metabolic side effects more common in olanzapine and clozapine, whereas anticholinergic side effects are more prominent in clozapine and quetiapine. However, apart from clozapine, the antipsychotic efficacy across pines is overall comparable [3]. On an individual level, however, significant differences occur [4]. A significant proportion of patients undergo more than one antipsychotic treatment before an effective treatment regimen is found. Thus, clinicians often have to assess whether a given antipsychotic treatment is insufficiently effective or has simply not achieved its effect yet [5]. In these situations, proper timing can be of clinical significance and presupposes a good understanding of when the onset of action can be expected.

There are however some inconsistencies between guidelines. The National Institute for Health and Care Excellence (NICE) recommends the continuation of antipsychotic drug treatment at optimum dosage for 4–6 weeks before rejecting the treatment [6]. The World Federation of Societies of Biological Psychiatry (WFBSP) recommends a treatment trial at optimal dose for at least 2 weeks and no longer than 8 weeks [7]. The British Association of Psychiatry (BAP) questions the recommendation that an adequate antipsychotic trial should last for 4–6 weeks, but acknowledges that it could be premature to declare failure of an antipsychotic treatment after 2 weeks [8].

These inconsistencies reflect that our knowledge of how antipsychotic drug effect develops over time is limited, in terms of both time to onset of action and time to maximum effect. Several studies have indicated an onset of action within the first week of treatment and thereby questioned the widespread perception that antipsychotic effect occurs with a significant delay [9,10,11,12,13]. However, data on when the maximum effect at a given dose occurs are limited.

Retaining patients in ineffective antipsychotic treatment regimens and discontinuing potentially effective treatment regimens prematurely are equally unwanted. Thus, a better understanding of how antipsychotics exert their effect over time may optimize their clinical use.

The purpose of this study is to estimate the onset of action and time to maximum antipsychotic effect by means of reductions in psychotic symptom scores in a subgroup of second-generation antipsychotics that share pharmacodynamic similarities [2]: asenapine, clothiapine, clozapine, loxapine, olanzapine, quetiapine, and zotepine (pines).

## 2. Materials and Methods

This study was performed according to PRISMA-guidelines [14]. The review was not registered in PROSPERO, as the original intention was a student project. The protocol was not published.

### 2.1. Types of Studies

We included randomized, placebo-controlled trials published in peer-reviewed journals in English.

### 2.2. Types of Participants

Adults ≥18 years of age diagnosed within the schizophrenic spectrum equivalent to F2 in the International Classification of Diseases, 10th Edition (ICD-10).

### 2.3. Types of Interventions

Treatment with an SGA of the pine group, defined according to the WHO classification system for Anatomical Therapeutic Chemicals (ATC-groups) N05AH (asenapine, clothiapine, clozapine, loxapine, olanzapine, and quetiapine) and N05AX (zotepine).

A placebo-arm was required as control. Studies were excluded, if participants were allowed to receive more than one fixed antipsychotic prescription during the trial.

### 2.4. Type of Outcome

Only studies using at least one of the following psychotic symptom scores were included: The Positive and Negative Syndrome Scale (PANSS) [15], the Scale for the Assessment of Positive Symptoms (SAPS) [16], the Brief Psychiatric Rating Scale (BPRS) [17,18,19], or Clinical Global Impression (CGI) [20]. Studies were required to assess one of these rating scales at baseline and at least once during the intervention period. Only studies with a follow-up time of at least one week after baseline assessment were included.

When more than one assessment score was available, the following hierarchy order was used: PANSS > SAPS > BPRS > CGI.

The primary outcome was the change in total symptom score from baseline to each of the subsequent follow-up time points for the intervention group and the placebo group respectively (Figure 1).

### 2.5. Search Method for Identification of Studies

We performed a systematic literature search of the electronic databases MEDLINE/PubMed, Embase, PsycInfo, and Cochrane Central Register of Controlled Trials (CENTRAL). The databases were searched from date of creation until 16 August 2022.

Entries for antipsychotics of the -pine group were combined with entries for schizophrenia, psychosis, or related illnesses, and entries for placebo treatment. The MEDLINE/PubMed and Embase searches were limited to randomized clinical trials using validated search filters [21]. No further limitations were applied. We defined pine-antipsychotics according to the WHO classification system for Anatomical Therapeutic Chemicals (ATC). We searched for pine-antipsychotics belonging to the ATC-groups N05AH (asenapine, clotiapine, clozapine, loxapine, olanzapine, and quetiapine) and N05AX (zotepine). Links to the complete search strings are included in the Appendix A.

### 2.6. Study Selection

All entries were screened for title and abstract. Studies not excluded based on title and abstract were reviewed in full text. Three authors (R.M., I.D., or K.S.) performed both screening for title and abstract and review of full texts independently. Discrepancies were resolved by agreement between two authors (R.M. and K.S.).

### 2.7. Data Extraction and Management

Data were extracted by two authors (R.M. and K.S.). Discrepancies were resolved by consensus.

Outcome data were extracted as mean PANSS, SAPS, BPRS, or CGI at baseline and at least one follow-up time point for both placebo and intervention group. Data for all available follow-up time points from week 1 onwards were recorded. When data were presented graphically, numbers were extracted digitally using the WebPlotDigitizer [22]. Mixed-effect Model Repeated Measures (MMRM) were preferred over Last Observation Carried Forward (LOCF) when both methods for handling missing data were available.

When studies had multiple study-arms with different dose regimens, only the arm with the highest dose that fell within prescribing information from U.S. Food and Drug Administration (FDA) and product information from the European Medicines Agency (EMA) [23,24] was included. This was done to ensure sufficient antipsychotic dose regimens but also to avoid a unit-of-analysis error due to repetitive entries of the same placebo group (details about all included studies occur in the Appendix A).

Variance was extracted as SD or SE for each data point. Alternately, SE or *p*-values for the difference between intervention and placebo groups were used to calculate variance using the built-in calculator in the Review Manager 5.3 software. If variance was given for one data-point only, this value was extrapolated to all data points within the same study. In one case [25], SE at week 1 was extrapolated from another study’s week 1 SE value [26].

For all included studies, the following study characteristics were extracted: Number of patients, mean age, percentage of male participants, and mean symptom score at baseline. Furthermore, we recorded whether the population was enriched, i.e., only included treatment responders, and whether placebo-responders were withdrawn. Finally, we registered whether a study contained only one follow-up measure or sequential measures. Unless explicitly stated, studies were registered as non-enriched population and not withdrawing placebo responders.

### 2.8. Assessment of Risk of Bias

The Revised Cochrane Risk of Bias Tool 2 was used to evaluate the quality of each of the included studies [27]. Two authors (R.M. and I.D.) evaluated all studies independently. Discrepancies were resolved by consensus.

### 2.9. Measure of Treatment Effect

The effect was measured as the difference in the changes in total symptom score from baseline to follow-up between intervention and placebo (Figure 1).

### 2.10. Synthesis of Results

Using the ReviewManager (RevMan). Version 5.3, The Nordic Cochrane Centre, 2014; Copenhagen, Denmark we performed pooled meta-analyses including all immediate-release antipsychotics and separate meta-analyses for each included antipsychotic and each formulation, as well as different time-point estimates (Figure 1). We used standardized mean difference (SMD), permitting comparisons between different symptom scores.

A Chi^2^-test was used to assess heterogeneity among studies [28]. In case of significant heterogeneity, Chi^2^ *p*-value < 5%, a random effects model was used. Otherwise, a fixed effects model was used.

### 2.11. Additional Analysis

To identify the possible impact of sequential vs. non-sequential data assessment, the possible impact of enriched vs. non-enriched populations, and the possible impact of systematic withdrawal of placebo-responders, we performed 3 sensitivity analyses and compared their results to the main analyses.

## 3. Results

### 3.1. Study Selection

Results of the literature search are illustrated in Figure 2. The search of the bibliographic databases MEDLINE/PubMed, Embase, PsycINFO, and Cochrane Central Register of Controlled Trials (CENTRAL) identified 4132 studies. After removal of duplicates, 2394 studies remained. 2267 studies were excluded based on screening of headlines and abstracts. The remaining 127 full-text studies were reviewed and 36 were found eligible for inclusion in this meta-analysis. For one study [29], data was inexplicable, and the authors were contacted but did not respond to our inquiry. Thus, *n* = 35 studies were included in the meta-analyses of this review: sublingual asenapine *n* = 4 [25,26,30,31], transdermal asenapine *n* = 1 [32], oral olanzapine *n* = 18 [31,33,34,35,36,37,38,39,40,41,42,43,44,45,46,47,48,49], long-acting injectable (LAI) olanzapine *n* = 1 [50], oral quetiapine *n* = 8 [51,52,53,54,55,56,57,58], extended-release (ER) oral quetiapine *n* = 3 [57,58,59], and oral zotepine *n* = 3 [60,61,62]. No studies investigating clothiapine, clozapine, or loxapine fulfilled the inclusion criteria due to either augmentation treatment, lack of placebo group, or incomplete outcome data.

### 3.2. Description of Included Studies

A detailed description of the included studies is presented in the Appendix A.

Thirty-five studies, published between 1995 and 2020 with *n* = 12,218 randomized patients, were included, of which *n* = 6331 patients were included in these analyses. The number of randomized patients in the included studies varied from 12 to 669. The mean age ranged from 34.2 to 42.3 and the range of male participants were 48.1–100% with a median percentage of 68.7%. Seven studies included data subgroup analyses for treatment interactions by age or gender [30,33,37,46,48,50,59]. One study found a statistically significant improvement in PANSS total score in patients <55 years compared to patients >55 years [46]. None of the studies found a treatment interaction by gender.

Apart from a diagnosis of schizophrenia, one study also included patients with schizoaffective disorder [54]. Twenty-six studies included patients with an acute exacerbation of psychotic symptoms [25,26,30,31,32,33,34,35,37,41,44,45,46,47,48,49,52,53,54,55,56,57,58,59,60,61]. The duration of the studies varied from 2 to 26 weeks: one study had a duration of 2 weeks [54], one study of 3 weeks [51], seven studies of 4 weeks [37,38,39,40,41,46,48], 21 studies of 6 weeks [25,26,30,31,32,33,34,35,42,43,44,45,47,49,52,53,55,56,57,58,59], three studies of 8 weeks [50,60,62], one study of 24 weeks [36] and one study of 26 weeks [61]. The number of participants treated with each pine and formulation emerges from Table 1.

In 27 studies, PANSS was recorded as primary outcome, and in eight studies, BPRS. The PANSS total score at baseline varied from 64.8 to 102.5, and the BPRS baseline score varied from 38.0 to 58.3. Two studies did not report a baseline score [26,41]. One study reported an erroneous baseline PANSS total score of 23.9, as PANSS consists of 30 items scored between 1 to 7, resulting in a PANSS total score in the range of 30 to 210 [43]. Still, these studies were included since the changes in symptom score from baseline to subsequent weeks were available.

12 studies used multiple study-arms investigating different antipsychotic dose regimens [26,30,31,32,36,42,47,50,53,56,57,58]. Table 2 shows the dosages chosen for inclusion in the meta-analyses.

Twenty-four studies included a series of consecutive follow-up time points, i.e., more than one follow-up time point (Forest plots for the sensitivity analyses are available in the Appendix A). The consecutive data of three studies were not used: one study only reported statistically significant results [56], and two studies reported consecutive data for observed cases without reporting the number of participants for each specific week [30,47]. For two studies, only consecutive data for the first 2 weeks were used as the study design allowed additional antipsychotic drugs in the subsequent weeks [54,55].

### 3.3. Risk of Bias in Included Studies

Risk of bias of each study included in the meta-analyses is presented in the Appendix A. The domain of missing outcome data was evaluated as high risk of bias in all studies. The high discontinuation rates and unbalanced dropout rates between groups are expected because participants drop out due to lack of efficacy, whereby missing data depends on the true value. In one study [51], only one participant in the placebo group discontinued, but since only four participants were randomized to the placebo group this was considered a significant proportion and thus a risk of bias.

Risk of bias in the measurement of the outcome are evaluated as “some concern” in 27 studies as the studies did not provide sufficient information on the training of personnel, ongoing validation of the use of the rating scales, and interrater reliability. In addition, there is a risk of diagnostic detection bias as the score of the outcome might reveal to the assessor which intervention group the participant is randomized to.

Even though several studies reported a trial registration number, we only found one study protocol [32] explaining a detailed analysis plan. All other studies were evaluated “some concern” for bias related to selection of reported results.

### 3.4. Meta-Analyses

We performed 53 separate meta-analyses. The number of meta-analyses performed for each pine, the respective follow-up time points, the number of included studies and participants, as well as standardized mean difference for each meta-analysis can be found in Table 1 (All Forest plots are available in the Appendix A).

We found a statistically significant difference between immediate-release pines and placebo at week 1 (SMD −0.20 [CI95% −0.28, −0.13]), and a trend towards an improved effect was found until week 3. No statistically significant improvements were found beyond that time point (Figure 3A).

We found the earliest statistically significant difference in symptom score between trial drug and placebo at the following time points:Week 1 for sublingual asenapine (SMD, −0.23 [CI95% −0.44, −0.02]) (Figure 3B), olanzapine (SMD −0.27 [CI95% −0.38, −0.17]) (Figure 3D), and zotepine (SMD −0.42 [CI95% −0.8, −0.03]) (Figure 3F);Week 2 for quetiapine (SMD −0.20 [CI95% −0.36, −0.05]) (Figure 3E), LAI olanzapine (SMD −0.36 [CI95% −0.65, −0.08]) (Figure 3G), and quetiapine ER (SMD −0.46 [CI95% −0.72, −0.20]) (Figure 3H);Week 3 for transdermal asenapine (SMD −0.21 [CI95% −0.41, −0.02]) (Figure 3C).

A tendency towards further improvement in symptom scores was found until the following time points:Week 4 for quetiapine (SMD −0.47 [CI95% −0.71, −0.24]) (Figure 3E);Week 5 for olanzapine (SMD −0.70 [CI95% −0.84, −0.57]) (Figure 3D) and quetiapine ER (SMD −0.86 [CI95% −1.12, −0.59]) (Figure 3H);Week 6 for sublingual asenapine (SMD −0.47 [CI95% −0.60, −0.33]) (Figure 3B), transdermal asenapine (SMD −0.30 [CI95% −0.50, −0.11]) (Figure 3C), and zotepine (SMD −0.77 [CI95% −1.17, −0.37]) (Figure 3F);Week 8 for olanzapine LAI (SMD −0.74 [CI95% −1.03, −0.45]) (Figure 3G).

However, no further statistically significant improvement was found beyond the earliest significant difference (Table 1 and Figure 3B–H).

One study presented data for differences between olanzapine and placebo at week 26 (SMD −0.12 [CI95% −0.53, 0.29]) and one study for the difference between zotepine and placebo at week 16 (SMD −0.40 [CI95% −0.76, −0.03]), week 20 (SMD −0.43 [CI95% −0.80, −0.07]), and week 26 (SMD −0.46 [CI95% −0.83, −0.10]) [36,61] (Table 1).

### 3.5. Additional Analyses

Twenty studies with sequential data, i.e., studies with more than one follow-up time point, were used for sensitivity analyses: sublingual asenapine *n* = 2 [25,26], oral olanzapine *n* = 9 [33,37,40,41,42,43,44,45,46], LAI olanzapine *n* = 1 [50], oral quetiapine *n* = 5 [51,52,53,54,55], ER oral quetiapine *n* = 1 [59], and oral zotepine *n* = 2 [60,61]. A sensitivity analysis solely including these studies was performed. The summary effect estimates did not differ significantly from the summary estimates of the main analyses for any of the included antipsychotics. Compared to the main analyses, more homogeneity was found for olanzapine at week 4 and significant heterogeneity was found for zotepine at week 8. Forest plots are available in the Appendix A.

Seventeen studies used an enriched population, i.e., only treatment responders are included: Sublingual asenapine *n* = 3 [25,30,31], transdermal asenapine *n* = 1 [32], oral olanzapine *n* = 11 [31,34,35,36,37,39,44,45,46,48,49], LAI olanzapine *n* = 0, oral quetiapine *n* = 3 [55,57,58], and ER oral quetiapine *n* = 2 [57,58]. A sensitivity analysis only showed statistically significant difference between enriched and non-enriched populations for quetiapine ER at week 6: (summary of SMD −0.41 [CI95% −0.61, −0.21]) and (summary of SMD −0.89 [CI95% −1.16, −0.62]). Data are presented in the Appendix A.

Nine studies reported exclusion of placebo-responders after the washout period: sublingual asenapine *n* = 5 [25,26,30,31], transdermal asenapine *n* = 1 [32], olanzapine *n* = 2 [42,47], quetiapine *n* = 1 [56], and zotepine *n* = 1 [62]. Only one of these studies [42] had sequential data. A sensitivity analysis was made by excluding these studies from the main meta-analyses. The sensitivity analysis showed no statistically significant difference from the main analyses. Data are presented in the Appendix A.

## 4. Discussion

This study estimates the onset of antipsychotic action as well as the time to maximum treatment response in pines, a subgroup of SGA that share pharmacological similarities.

The overall analysis, including all immediate-release pines, shows an early onset of antipsychotic action within the first week of treatment and a maximal response after 3 weeks of treatment. Not surprisingly, analyses of the individual pines show more variable results. While oral asenapine, olanzapine, and zotepine show an onset of antipsychotic action within 1 week, the effect of quetiapine, LAI olanzapine, and ER quetiapine does not differ significantly from placebo before 2 weeks of treatment and 3 weeks of treatment for transdermal asenapine. Across all included pines, we found a tendency towards small increasing effects up to 8 weeks after treatment initiation, which however was no longer statistically significant beyond week 3. The conclusiveness of these individual analyses is limited, as the amount of data is small and variability correspondingly increased compared to the overall analysis. Furthermore, the slight delayed onset of quetiapine can be explained by the fact that the dosage followed a titration regimen reaching target dose after approximately 7 days.

Our findings challenge treatment guidelines that recommend treatment at optimal dosage between 2 to 8 weeks before rejecting the treatment [6,7,8].

Our findings are consistent with several previous studies. A meta-analysis examined time to onset of action of chlorpromazine, haloperidol, risperidone, and olanzapine and found a significant difference in the percentage reduction in mean total symptom scores between treatment and placebo group after one week of treatment [9]. In addition, the analysis showed a significantly higher percentage reduction in mean total symptom score after the first 2 weeks of treatment than the subsequent 2 weeks (17.2% vs. 6.7% [*p* < 0.001]). The analysis was extended by including amisulpride and the reduction in symptom score in individual patient data was analyzed in order to reach a more homogeneous analysis [10]. The data were pooled, and confirmed the previous findings of a significantly larger reduction in symptom score during weeks one and two than during weeks three and four. A prospective study distinguished between patients being non-responders to risperidone and patients having an early onset of response defined as ≥20% improvement in PANSS total score from baseline to week two [13]. They found that early onset of response to risperidone was highly predictive for subsequent improvement in psychotic symptoms during a 10-week follow up. Another meta-analysis showed that 90% of the patients showing non-improvement (<20% reduction in PANSS or BPRS score) after 2 weeks of treatment did not show much improvement (<50% reduction in PANSS or BPRS score) at endpoint (4–12 weeks), and 55% did not even improve minimally (<20% reduction in PANSS or BPRS score) [11]. These results highlight lack of early effect as an important predictor for the overall efficacy that can be achieved in the individual patient.

A study found that 22.5% of patients with first-episode psychosis (FEP) did not respond to antipsychotic treatment before 4 weeks of treatment and 11.5% did not respond before 8 weeks [63]. Another study showed an increase in estimated cumulative treatment response rate from 39.6% at week 8 to 65.2% at week 16 [64].

However, none of these analyses aimed to investigate the time until the maximum effect is achieved.

Whether an antipsychotic treatment should be changed immediately or maintained for a longer period is a common clinical dilemma. Thus, our results add important information to existing knowledge.

Since our analysis included data from longitudinal studies with different follow-up time schedules, we conducted separate meta-analyses for each time point (Figure 1). We assumed that meta-analyses based on studies with numerous consecutive follow-up time points would produce a far more homogeneous estimate if they were not pooled with data from studies with only one follow-up time point, thereby potentially reducing the validity of our results. However, a sensitivity analysis that excluded studies with only one follow-up time point showed that this did not affect our summary estimates or change heterogeneity significantly, indicating a certain robustness.

It is not uncommon for psychiatric clinical trials to use an enriched population or withdraw placebo responders, in order to determine a drug’s optimal effect, minimize the discontinuation rates, or obtain a more homogenous population, and lower the number of patients needed to achieve a statistically significant treatment effect [65]. Both designs, however, hold a risk of overestimating the treatment effect. Therefore, we also performed sensitivity analyses excluding studies using enriched populations or withdrawing placebo responders. We found a statistically significant difference between the enriched and non-enriched population for quetiapine ER at week 6 while results for asenapine, olanzapine, and quetiapine remained unchanged. Although these analyses must be interpreted with caution due to the sparse amount of data, neither the inclusion of enriched populations nor withdrawal of placebo-responders appear to overestimate the effect of the included antipsychotics.

The pharmacokinetic properties of pines imply that steady state occurs within 1 week after treatment initiation. Since a homogeneous distribution to body compartments occurs within one week of treatment, the improvement in effect beyond this time point must be due to other factors. By visual inspection, the effect of extended-release pines has a tendency to show continuous improvement several weeks after treatment initiation. However, this cannot be explained by differences in steady state, as an extended-release formulation share half-life with the immediate-release formulation.

Clinical guidelines recommend treatment with antipsychotics at optimal doses for at least two weeks and no longer than eight weeks [6,7,8]. In addition, it is acknowledged that it may be premature to switch an ongoing treatment after only 2 weeks. Although our results reflect changes in symptom scores on a population level and does not predict the individual course of treatment, our findings support that an onset of antipsychotic effect appears within two weeks of treatment. Thus, a total lack of effect behind this timeframe should lead to termination of an ongoing antipsychotic treatment unless individual considerations contraindicate this.

Our study has limitations.

First, some data points are based on sparse data.

Second, most pines have only been studied in short term studies and shows in the individual pines no clear stabilization of the response rate towards the end of the studied time period. Thus, a progress in response beyond 6 weeks cannot be excluded.

Third, some studies only included patients who did not exceed an upper symptom score. Consequently, our findings may not apply to severely ill patients.

Fourth, differences in length of washout periods for antipsychotics prior to inclusion might influence the magnitude of symptom scores, as both onset of antipsychotic action and time to maximal effect rely on the systemic response caused by the earlier dopamine antagonistic agent on the brain and the length of time for washout [9]. However, this mechanism also applies to the placebo group and therefore does not influence the difference between response to placebo and antipsychotic drug.

Fifth, previous studies have shown a difference in time until response of antipsychotic treatment between patients with multi-episode psychosis and FEP. As the included studies in our meta-analyses only investigated multi-episode psychosis, the results should not be generalized to patients with an FEP.

Sixth, we included antipsychotics with similar multi-acting receptor-targeting profiles (pines) and our findings are confined to those and are not necessarily generalizable to all antipsychotics. However, our results are in line with data from previous meta-analyses including other antipsychotics.

Seventh, several of the included studies addressed missing data using Last Observation Carried Forward. This contributes bias as it underestimates the effect of the treatment.

Finally, our estimates are solely based on psychometric measures. Patient-centered, subjective outcomes are left out.

## 5. Conclusions

The present study shows that the onset of effect of immediate-release antipsychotics of the pine group is obtained within the first week of treatment. Furthermore, their effect does not increase beyond 3 weeks of treatment. Separate analyses of the individual pines, including modified-release formulations of olanzapine and quetiapine, show a more variable outcome, which, however, due to sparse data, is difficult to draw conclusions. Hence, trials with an extended duration would be of relevance.

## Figures and Tables

**Figure 1 biomedicines-11-00082-f001:**
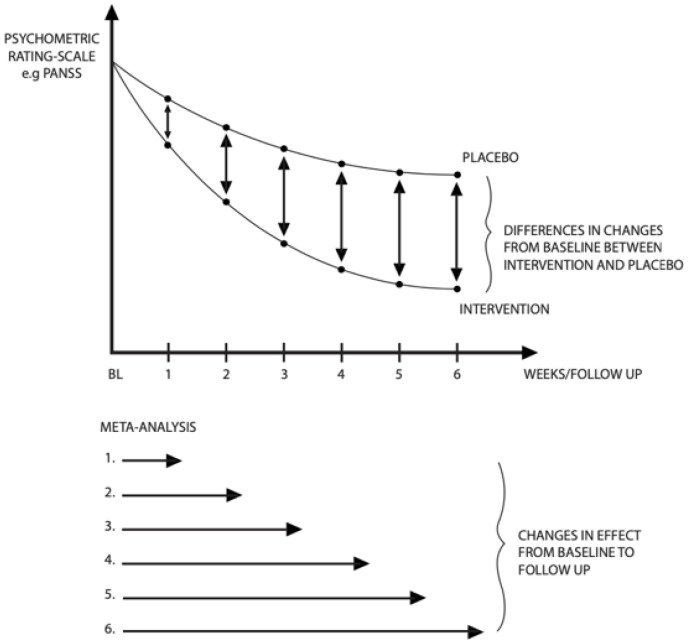
The top figure illustrates the difference in the changes of symptoms score from baseline to follow-up between intervention and placebo. The lower figure illustrates that separate meta-analyses were made for each week.

**Figure 2 biomedicines-11-00082-f002:**
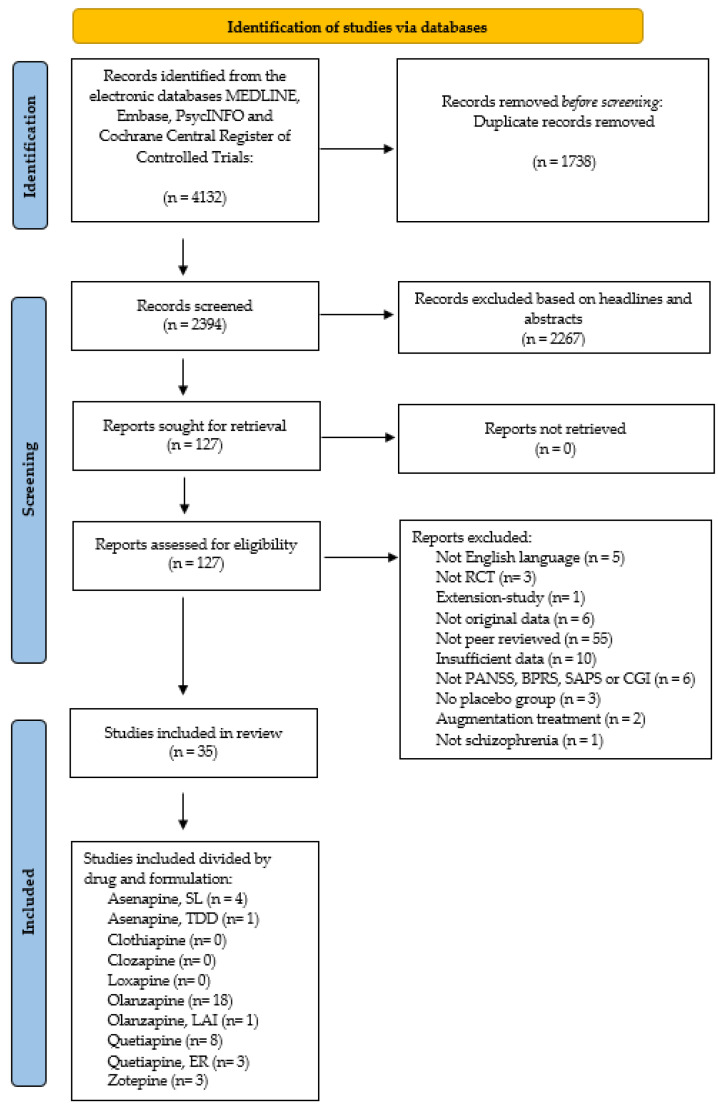
Illustrates the PRISMA flowchart of the study selection process [14]. Since 3 studies [31,57,58] investigated 2 antipsychotics of the -pine group, the sum of studies divided by drug does not correspond to the total number of studies included in the review. [ER = extended-release, LAI = long-acting injectable, SL = sublingual, TDD = transdermal drug delivery].

**Figure 3 biomedicines-11-00082-f003:**
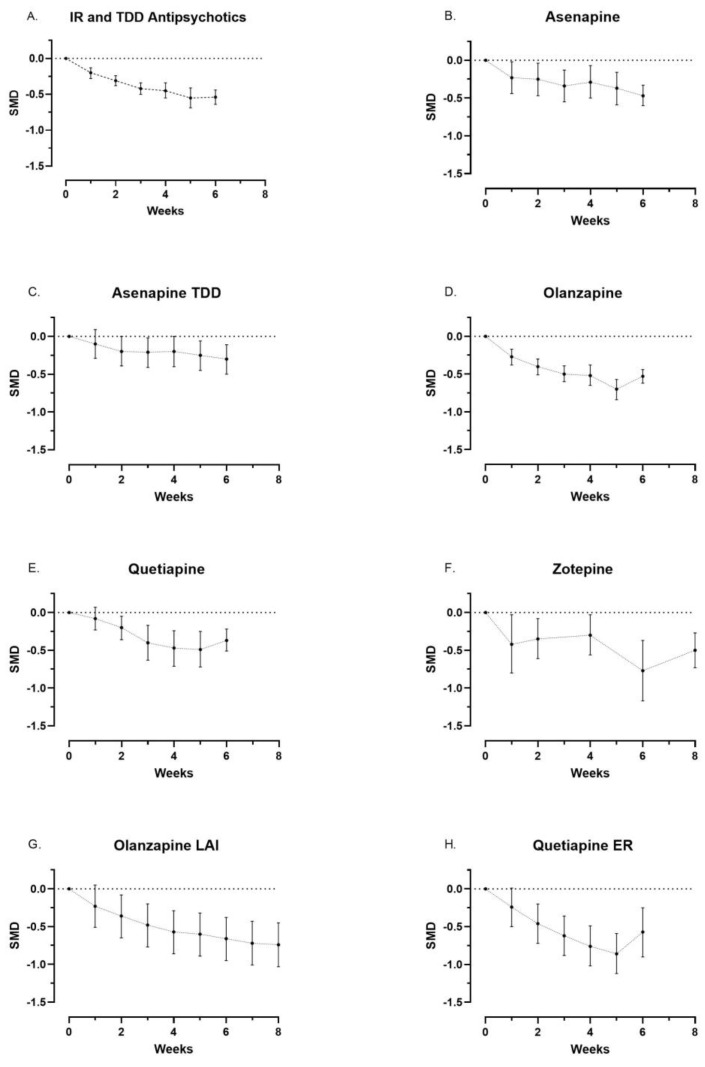
(**A**–**H**) Illustrates the standardized mean difference (SMD) in the changes in symptom score from baseline to follow-up between intervention and placebo (dots) with 95% confidence intervals (bars). The data also emerges from Table 1 and the Forest plots in the Appendix A. [ER = extended-release, LAI = long-acting injectable, TDD = transdermal drug delivery].

**Table 1 biomedicines-11-00082-t001:** Shows the number of meta-analyses made for each pine, and each of their follow-up time point with corresponding summary estimate (SMD [CI95%]).

SGA ^1^	Number of Meta-Analyses	Follow-Up Time Points for the Meta-Analyses, Weeks after Baseline	Number of Studies	Number of Participants	Summary Estimate (SMD ^1^ [CI95%])
Immediate release and transdermal drug delivery	6	1	18	3068	−0.20 [−0.28, −0.13]
		2	19	3125	−0.31 [−0.38, −0.24]
		3	15	2508	−0.42 [−0.50, −0.34]
		4	19	3105	−0.45 [−0.55, −0.34]
		5	11	1960	−0.55 [−0.69, −0.41]
		6	22	4097	−0.53 [−0.62, −0.44]
Asenapine	6	1	2	345	−0.23 [−0.44, −0.02]
		2	2	345	−0.25 [−0.47, −0.04]
		3	2	345	−0.34 [−0.55, −0.13]
		4	2	345	−0.29 [−0.50, −0.07]
		5	2	345	−0.37 [−0.59, −0.16]
		6	4	907	−0.47 [−0.60, −0.33]
Asenapine, TDD ^1^	6	1	1	406	−0.10 [−0.29, 0.09]
		2	1	406	−0.20 [−0.39, −0.00]
		3	1	406	−0.21 [−0.41, −0.02]
		4	1	406	−0.20 [−0.40, −0.00]
		5	1	406	−0.25 [−0.45, −0.06]
		6	1	406	−0.30 [−0.50, −0.11]
Olanzapine	7	1	9	1466	−0.27 [−0.38, −0.17]
		2	9	1460	−0.40 [−0.51, −0.30]
		3	9	1455	−0.50 [−0.60, −0.39]
		4	12	1839	−0.52 [−0.65, −0.38]
		5	6	919	−0.70 [−0.84, −0.57]
		6	11	1892	−0.66 [−0.76, −0.57]
		26	1	104	−0.12 [−0.53, 0.29]
Quetiapine	6	1	5	745	−0.08 [−0.23, 0.07]
		2	5	689	−0.20 [−0.36, −0.05]
		3	3	302	−0.40 [−0.63, −0.17]
		4	2	290	−0.47 [−0.71, −0.24]
		5	2	290	−0.49 [−0.72, −0.25]
		6	5	786	−0.37 [−0.51, −0.22]
Zotepine	8	1	1	106	−0.42 [−0.80, −0.03]
		2	2	225	−0.35 [−0.61, −0.08]
		4	2	225	−0.30 [−0.56, −0.03]
		6	1	106	−0.77 [−1.17, −0.37]
		8	3	304	−0.50 [−0.73, −0.27]
		16	1	119	−0.40 [−0.76, −0.03]
		20	1	119	−0.43 [−0.80, −0.07]
		26	1	119	−0.46 [−0.83, −0.10]
Olanzapine, LAI ^1^	8	1	1	196	−0.23 [−0.51, 0.05]
		2	1	196	−0.36 [−0.65, −0.08]
		3	1	196	−0.48 [−0.77, −0.20]
		4	1	196	−0.57 [−0.86, −0.29]
		5	1	196	−0.60 [−0.89, −0.32]
		6	1	196	−0.66 [−0.95, −0.38]
		7	1	196	−0.72 [−1.01, −0.43]
		8	1	196	−0.74 [−1.03, −0.45]
Quetiapine, ER ^1^	6	1	1	236	−0.24 [−0.50, 0.01]
		2	1	236	−0.46 [−0.72, −0.20]
		3	1	236	−0.62 [−0.88, −0.36]
		4	1	236	−0.76 [−1.02, −0.49]
		5	1	236	−0.86 [−1.12, −0.59]
		6	3	631	−0.57 [−0.90, −0.25]

^1^ ER= extended-release, LAI = long-acting injectable, TDD = transdermal drug delivery, SGA = Second Generation Antipsychotics, SMD = Standardized mean difference.

**Table 2 biomedicines-11-00082-t002:** Some of the included studies had multiple study-arms investigating different dose regimens. The first column shows the dose regimens investigated for different pines. The second column shows the dose regimen selected for inclusion in our meta-analyses.

SGA ^1^	Dose Regimens Investigated in Included Studies, mg	Chosen Dose Regimens for Included Studies, mg
Asenapine, *n* = 4 [25,26,30,31]	5 10 20	10–20
Asenapine, TDD ^1^, *n* = 1 [32]	3.8 7.6	7.6
Olanzapine, *n* = 18 [31,33,34,35,36,37,38,39,40,41,42,43,44,45,46,47,48,49]	1 5 2.5; 5; 7.5 (flexible dosing) 10 7.5; 10; 12.5 (flexible dosing) 15 12.5; 15; 17.5 (flexible dosing) 10–20 (flexible dosing) 20	10–20
Olanzapine LAI ^1^, *n* = 1 [50]	210/2 weeks 300/2 weeks 405/4 weeks	300/2 weeks
Quetiapine, *n* = 8 [51,52,53,54,55,56,57,58]	75 150 Max 250 mg (flexible dosing) 250 300 400 600 750 Max 750 mg (flexible dosing) 600; 800 (flexible dosing)	Max 250–800
Quetiapine ER ^1^, *n* = 3 [57,58,59]	300 400 600 800	600–800
Zotepine, *n* = 3 [60,61,62]	Max 225 (flexible dosing) 300	Max 225–300

^1^ ER = extended-release, LAI = long-acting injectable, SGA = Second Generation Antipsychotics, TDD = transdermal drug delivery.

## Data Availability

The data presented in this study are available in the Appendix A.

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
