# Peer review of "Onset of Action of Selected Second-Generation Antipsychotics (Pines)–A Systematic Review and Meta-Analyses"

_biomedicines, 2022, doi:10.3390/biomedicines11010082_

Round 1

Reviewer 1 Report

Dear Authors,

the  addressed topic is highly intersting in psychiatry. The manuscript  is well organized and written. Tables are appreciable. Limitations of the study are clearly stated.

Therefore, I would solely suggest minor revisions: 

Basic explanation about known pharmacodynamics and/or pharmacokinetics  for different "pines" could help understanding the presented results.

Letters in figure 3 should be clearly mentinoned in the main text when describing single drugs.

Kind regards

Reviewer 2 Report

The manuscript from Meyer et al. performed a database survey to estimate the onset and time to maximise the effects of second-generation antipsychotics, which involved more than 12,000 individuals, including 6,000+ patients. Their findings generally agree with the established concepts and models that, whilst noticeable patterns exist between different pine families, a higher reduction in gross symptom occur within or immediately after the first two weeks of treatment. For the purpose of providing a source of summarised data and associated analysis, I would recommend the manuscript appropriate be published for the convenience of the community.

Reviewer 3 Report

From the clinicians' point of view, the study conducted is very relevant. However, there is little clarity in the way the results are presented. The following is a list of comments:

Abstract - (lines 16-17) only the drugs actually included in the meta-analysis should be listed here. Listing all of them is misleading to the potential reader, who will not find information on all the drugs, but only on some of them in the rest of the article. As the authors rightly did, they described the process of searching the databases (3.1.), and only here should it be stated that information on studies concerning the other drugs in this group was also searched.

All study results are included in the supplementary materials. This is inconvenient for the person analysing the text, as he/she has to wade through the entire, uniform text in search of the information of interest.

As I mentioned above, the research results should be found in a stabelled form in the main text. 

Section 3.2 characterises the populations studied, but nothing comes of it. There is no information on whether the studies analysed describe the effect of, for example, gender or age on the time to onset of patient improvement. If such information appears, this should be indicated. If not and no studies refer to demographics, this should also be indicated and described in the conclusions. 
